

# Multi-annual variability of a new proxy-constrained modeled AMOC from 1450-1780 CE

Eric Samakinwa[1,2], Christoph C. Raible[2,3], Ralf Hand[1,2], Andrew R. Friedman[1,2], and Stefan Brönnimann[1,2]

[1]Institute of Geography, University of Bern, Switzerland.
[2]Oeschger Center for Climate Change Research, University of Bern, Switzerland.
[3]Climate and Environmental Physics, Physics Institute, University of Bern, Bern, Switzerland.

**Correspondence:** Eric Samakinwa (eric.samakinwa@unibe.ch)

**Abstract.**

The ongoing discussion about the Atlantic Meridional Overturning Circulation (AMOC) slowdown over the 21[st] century requires an understanding of preindustrial AMOC variability. Here, we present a new AMOC reconstruction during the Little Ice Age from 1450–1780 CE, generated with a novel nudging technique. The technique uses a 10-member ensemble of
5  ocean-model simulations nudged to proxy-reconstructed sea surface temperature. The new reconstruction improves existing knowledge of the AMOC variability and underlying processes, showing that the AMOC weak phases under stable atmospheric $CO_2$ conditions are mainly driven by a 4-to-7-year lagged effect of surface heat flux associated with the North Atlantic Oscillation (NAO). However, the strong phases are a response to instantaneous surface wind stress. Analyses of our AMOC reconstruction reaffirm previous findings about the mechanisms of AMOC variability and its link to the NAO. In addition,
10  we show that processes leading to the phases of weak and strong AMOC are non-symmetric. Overall, the nudging technique enables us to better constrain past AMOC variability.





## 1 Introduction

The Atlantic Meridional Overturning Circulation (AMOC) has profound impacts not only on the climate of the North Atlantic but also on the global scale. It plays a significant role in redistributing heat and freshwater around the globe (Huhlbrodt et al., 2007; Buckley et al., 2016; Zhang et al., 2019). Many studies suggest a weakening of the AMOC in the $21^{st}$ century (Caesar et al., 2018, 2021) due to increased radiative forcing through anthropogenic activities. However, this remains questionable as recent studies have shown contrasting results attributing the AMOC changes to natural variability (Latif et al., 2022) and poorly constrained AMOC strength resulting from discrepancies across different paleo-proxy reconstructions (Latif et al., 2022; Kilbourne et al., 2022). Analysing past climate could help, but our knowledge of the AMOC variability before the instrumental era is still limited.

The RAPID AMOC measurement campaign began in 2004, providing real-time monthly observations of the stream function of the ocean (Frajka-Williams et al., 2019). As these observations exhibit large inter-annual variability (Srokosz et al., 2015), the record is still too short to infer a slowdown of the AMOC as suggested by modeling studies (Caesar et al., 2018, 2021). To overcome the length issue, proxy reconstructions of the deep ocean using carbon-isotope $\delta^{13}$C and $\delta^{18}$O from benthic foraminifera in sediment core are available at a few marine proxy locations in the North Atlantic (Bond et al., 2001; Oppo et al., 2003), but they are insufficient to fully resolve past AMOC variability. As more marine proxy reconstructions of more oceanic parameters from new archives, such as bivalve and Coralline Algae holding proxies such as Alkenones, Planktonic foraminifera and so on (Moffa-Sánchez et al., 2019; Kilbourne et al., 2022) begin to be unveiled, the spatiotemporal resolution typically favors the modern era, leaving us with inadequate knowledge to attribute changes in the AMOC to either natural or external forcing.

Moreover, indirect indices based on related surface fields, such as reconstructed sea surface temperature (SST) (Rahmstorf et al., 2015), are used to study AMOC variability. This technique uses SST in the subpolar gyre region to indicate changes in AMOC variability. Attributing the AMOC to surface variables in specific regions of the North Atlantic can be unreliable as changes in the SST field can result from multiple factors, and the influence of these factors can change over time (Latif et al., 2022; Kilbourne et al., 2022), implying a need for a 3-dimensional characterization of the AMOC. Coupled Global Climate Models (GCMs) help in providing the 3-dimensional state of the ocean, but historical AMOC transport changes show large variations across CMIP6 models (Weijer et al., 2020). The projected global surface temperature sensitivity to the doubling of preindustrial $CO_2$ ranges from 1.8 – 5.6 K across CMIP6 models (Zelinka et al., 2020), casting doubts on these models' uncertainty, which is strongly related to their projected AMOC response (Bonnet et al., 2021; Bellomo et al., 2021).

To overcome these deficiencies and uncertainties, proxy records and model simulations can be combined to produce a 3-dimensional proxy-constrained AMOC simulation under stable atmospheric $CO_2$. Such a technique enables a better specification of initial conditions for the hindcasts, forecasts, and projections of future changes. Furthermore, we can investigate the interplay between the AMOC and the North Atlantic Oscillation NAO, both of which have a non-linear behavior. Their responses can be sensitive to small changes in initial conditions or external forcing. Under stable $CO_2$ conditions, the relationship between the AMOC and NAO might exhibit different sensitivities and feedbacks compared to scenarios with changing





$CO_2$ concentrations. This non-linearity can result in contrasting patterns of interaction between the AMOC and NAO.

Here, we present a novel approach that goes beyond the classical method of simulating the preindustrial period, where the prescribed $CO_2$ is usually 280 ppmv, by nudging the simulated ocean surface temperature to an ensemble reconstruction of spatially-complete SST fields (Samakinwa et al., 2021) from 1450 - 1780 CE. We concentrate on this period as greenhouse

gases, in particular $CO_2$, are relatively stable (Meinshausen et al., 2017), and thus the assessment of AMOC variability under rather stable greenhouse gas concentrations is possible. This enables us to assess the major mechanisms operating during the different phases of the AMOC (Hofer et al., 2011; Haskins et al., 2020; Megann et al., 2021; Chafik et al., 2022) under rather unperturbed climate conditions and deliver a baseline understanding for any future change. We use an SST reconstruction that combines information from paleoclimate proxies with climate model simulations (Neukom et al., 2019), thereby provid-

ing physically consistent fields. We also use selected atmospheric variables from the ECHAM6 atmospheric model (Hand et al., 2023), which uses the same SST reconstruction as its boundary condition, thereby completing a unique setup capable of constraining past AMOC variability.

## 2  Data and Methods

### 2.1  Model description MPI-OM

The ocean model MPI-OM (Marsland et al., 2003) is an ocean general circulation model and a successor of the Hamburg Ocean Primitive Equation Model. The model dynamics are solved on an orthogonal, curvilinear C-grid. It has a formal resolution of $3.0° \times 1.8°$ (GR30) and spreads over 40 unequally spaced $z$-coordinate model levels. It also uses a dynamic-thermodynamic sea ice model of Hibler (1979) (Hibler, 1979) that simulates sea ice thickness and its distribution independence of ambient climatic conditions. Furthermore, overturning by convection is parameterized via increased vertical diffusion, and an isopycnal

scheme is applied for subgrid-scale mixing (Marsland et al., 2003).

### 2.2  Sea surface temperature reconstruction

The SST reconstruction depends on an existing ensemble of annual multi-proxy temperature reconstructions ($N_{y,i}$) (Neukom et al., 2019), and monthly SST observations in the recent period using HadISST2. The annual reconstruction is augmented with intra-annual and subgrid-scale variability using a multiple linear regression approach to give an estimate of SSTs at each

grid point (Samakinwa et al., 2021). We employ three oceanic indices calculated from the annual reconstruction in training the linear regression models, namely the Nino3.4 in the Pacific, Dipole Mode Index in the Indian Ocean, and the Tropical Atlantic SST Index in the Atlantic, to ensure consistency across basins. We utilize an additive time series model by decomposing HadISST2 into 4 different components, comprising of constant and varying parts (Equation 1). $C_i$ denotes the long-term mean SST over grid box $i$, which is constant for the reconstruction period, and $A_{m,i}$ represents the seasonal cycle which is regular

and predictable change per month $m$, that recur every calendar year, to complete the constant part. The varying part of the model consists of $T_{y,i}$ and $I_{m_y,i}$, with the former showing annual mean anomaly, while the latter shows monthly changes with





respect to years termed intra-annual variability. $I_{m_y,i}$ will then average to zero for each grid box. This is imperative since the desired annual mean is preserved in $N_{y,i}$.

$$HadISST_{y,m,i} = C_i + A_{m,i} + T_{y,i} + I_{m_y,i} \qquad (1)$$

Following this decomposition, $I_{m_y,i}$ is regressed on indices of the corresponding year, resulting in monthly stratified grid box linear models, such that;

$$I_{m_y,i} = \beta_{1(m,i)}Nino3.4_y + \beta_{2(m,i)}DMI_y + \beta_{3(m,i)}TASI_y \qquad (2)$$

where $\beta_{1(m,i)}$, $\beta_{2(m,i)}$ and $\beta_{3(m,i)}$ are monthly regression coefficients per grid point with response to annual Nino3.4, DMI, and TASI, respectively.

## 85  2.3  Atmospheric variables

The atmospheric variables used in this study are output from the historical simulations for Modern Era Simulations (ModE-Sim) (Hand et al., 2023). ModE-Sim utilizes the ECHAM6 (T63L47) atmospheric general circulation model (Stevens et al., 2013) and has been used as atmospheric background state in a Bayesian reconstruction framework and has demonstrated robustness in representing past climate events (Reichen et al., 2022). Boundary and initial conditions used in the simulations

follow the *past1000* experiments of PMIP4 (Hand et al., 2023) but with few exceptions. SST and Sea Ice Concentration (SIC) are prescribed from a proxy-observation-based reconstruction (Samakinwa et al., 2021). Furthermore, all forcings including land surface, volcanic aerosols, tropospheric aerosols, and greenhouse gas concentrations are specified following the PMIP4 protocol. Selected atmospheric variables utilized in this study include; dew point temperature, east-west stress, north-south stress, scalar wind, total precipitation, 2m-temperature, mean sea level pressure, total cloud cover, and total solar radiation.

These atmospheric variables are spatially and temporally interpolated to the model grid and time step.

## 2.4  MPI-OM Integration and Nudging

We spin up our simulations for 1419 years using atmospheric variables of a non-volcanic year in the previously described ModE-sim, while the deep ocean is allowed to evolve from an MPIOM pre-calculated 3-dimensional climatology (Levitus et al., 1998). Due to the long-memory nature of the ocean (Brönnimann et al., 2019), using a volcanic year will increase the

spin-up time considerably. Similarly, initializing from the pre-calculated 3-dimensional climatology provides the advantage of reducing the spin-up time for the deep ocean to reach a quasi-steady state against starting from an unknown condition and has proven to produce consistent variables in recent ocean model simulations (Samakinwa et al., 2020).

Furthermore, this study focuses on AMOC variability in which its index is usually estimated between 500 and 1500 m and polewards from 20°N in the North Atlantic hence it is imperative to achieve equilibrium in the deep, especially from the surface

down to a depth of 1525 m. Analyses of ocean heat content (OHC) for different depths of the ocean show that our spin-up procedure achieved a quasi-steady state on the ocean surface faster. Nudging ocean model at the surface to SST has proven to be effective in reconstructing near-past AMOC variability on different time scales ranging from interannual to multidecadal





(Ortega et al., 2017). The steady state at the sea surface results from the surface nudging of SST (Samakinwa et al., 2021) with a constant relaxation timescale of one year. In our implementation of nudging, a standard approach is applied on an annual
timescale by adding a latitudinal fixed restoring term to the equation conserving SST in the Ocean model (Equation 3).

$$\frac{\partial SST}{\partial t} = ... + \frac{\gamma}{\rho_o C_p h}(SST_{model} - SST_{recons}) \tag{3}$$

where $SST_{recons}$ is the reconstructed SST used for the surface nudging of MPIOM, $SST_{model}$ is the model simulated SST, $\rho_o$ is the ocean mean density, $C_p$ is the specific heat of the ocean, $h$ is the depth of the surface layer, while $\gamma$ is arbitrary and tun-able relaxation term that dynamically adjusts the strength of the nudging. In this setup, $\gamma$ corresponds to the relaxation time
scale. This minimizes biases in regions of strong temperature gradients such as the Gulf Stream in the North Atlantic, and also in regions of strong model biases where $SST_{model}$ could be significantly larger than the $SST_{recons}$ (Equation 4) such that;

$$SST_{model} - SST_{recons}\| \gg 0 \tag{4}$$

Also, the ability of the model to simulate realistically, the internal variability is crucial in achieving physically consistent
oceanic variables. The Ocean model MPIOM has been shown to simulate reasonable internal variability across different times-lices (Samakinwa et al., 2020). To complete our complex setup, we allow SST to vary between 60°N and 60°S according to the latitudinal averaged SST and decrease linearly to zero from 60°N/S polewards. While the SST varies from one year to the other according to the reconstruction, we implement surface-restoring salinity to climatological values from Steele et al. (2001) (Steele et al., 2001). In the deep ocean, a quasi-steady state is reached at different times depending on depth. At a depth of 420
m, a quasi-steady state is attained after 390 years, while at 825, 1220, and 1525 m, the corresponding equilibrium years are 1000, 1190, and 1350, respectively (Fig. S1). We carried out a 10-member ensemble of ocean simulations for the period 1450 - 1780, to identify the mechanisms of AMOC variability under stable atmospheric $CO_2$ concentration.

## 2.5 AMOC Phases and Variable Analyses

Selected atmospheric and oceanic variables are analyzed to infer the leading patterns and mechanisms influencing the strong
and weak phases of the AMOC. We select High AMOC index years $HAI_{year}$ as those in which the AMOC index exceeds +1 standard deviation SD, while years with lower than -1 SD are considered Low AMOC index years $LAI_{year}$. Normal AMOC index years $NAI_{year}$ are defined as years between -1 and +1(Fig. S2a). The Kolmogorov-Smirnov test (KS-test) is used to confirm that values greater than +1 SD and lower than -1 SD are from a distinct distribution from the normal. The probability density distribution following these thresholds shows that the distributions are well separated (Fig. S2b). This
approach distinguishes changes in the statistical properties of the AMOC time series by also considering the years between -1 and +1 as Normal AMOC index years $NAI_{year}$. With these selection criteria, we can sample 428 $HAI_{year}$ and 329 $LAI_{year}$, while the remaining part of the sample falls under the $NAI_{year}$ (N = 2553). Composites of the identified dominant atmospheric variables based on the $HAI_{year}$ (strong phases) and $LAI_{year}$ (weak phases) are analyzed relative to the $NAI_{year}$.



## 3 Results

### 3.1 Simulated mean state & variability of the AMOC

To evaluate our reconstruction approach, firstly, we compare the simulated AMOC at 26°N to proxy-based estimates of the AMOC. We found an agreement in the temporal structure between our simulation and an estimate of the AMOC strength based on the $\delta^{18}O$ in benthic foraminifera from sediment cores (Fig. 1) retrieved from the Laurentian Channel (Thibodeau et al., 2018). Similarly, we find a correlation of 0.35 between SST-based AMOC index (K) reconstruction (Rahmstorf et al., 2015) and the 10-year running average of our simulated SST ensemble mean (Fig. S3). We also compare the mean state and variability with existing AMOC studies. The simulated strength of the AMOC is defined as the maximum overturning stream function at 30°N (Song et al., 2019).

In our simulations, the AMOC strength is more evident when comparing seasonal averages, such as maximum overturning stream function over summer (JJA), and winter (JFM), than in the annual averages. While the strength of the AMOC reaches 25.8 Sv in both seasons, the corresponding value for the annual is 23.4 Sv (Fig. 1). Seasonal means are more pronounced than the annual mean, this is because the maxima in summer and winter are at different locations. Also, the inflow of deep water from the South Atlantic, the Antarctic Bottom Water (AABW), becomes weaker as it flows northward during winter. The loss of density of the AABW along its path is likely due to the bottom-intensified mixing by internal tides (de Lavergne et al., 2016; Vic et al., 2019) that are partly driven by surface wind stress. In general, the simulated pattern of the AMOC is physically consistent with outputs from coupled GCM simulations (Yoshimori et al., 2010; Jungclaus et al., 2013; Jackson et al., 2015; Liu et al., 2020; Haskins et al., 2020; Bellomo et al., 2021), where most clockwise circulations occur at depths shallower than 3000 m. We also compare seasonal (JFM & JJA) and annual variability of the AMOC using the standard deviation (Fig. 1). The highest standard deviations of 12 to 14 Sv are around the equator for the annual timescale. The standard deviation decreases away from the equator up to 60°N but still shows a range of 3 – 4 Sv. The structure of the AMOC variability is consistent across models but varies with resolution. Still, it is consistent with the AMOC standard deviation obtained from models of similar resolution (Hirschi et al., 2020). On the seasonal scales, the structure of variability in winter remains broadly similar to the annual variability, while it is strongly reduced in summer. Lastly, we evaluate our nudging scheme by simulating the AMOC with time-varying heat and freshwater fluxes but by restoring SST to its climatological values *no_nudge*, and comparing the results with the SST nudged *Nudged* simulations, where SST varies per year. We also compare the leading Empirical Orthogonal Function (EOF1) of both simulations with outputs from coupled GCMs. Our *Nudged* experiment shows an agreement in spatial structure with historical simulations of the CMIP6 models (Latif et al., 2022; Jungclaus et al., 2013) and an explained variance of 25.2%, while the *no_nudge* experiment misrepresents the pattern of variability and with an explained variance of 9.1% (Fig. S4). In the remaining part of this study, we focus on the *Nudged* experiments.





**Figure 1. Comparison of our simulated AMOC with reconstruction and outputs from coupled Global Climate Models.** The top panel shows the simulated annual mean AMOC (Sv) at 26°N and AMOC proxy reconstruction based on $\delta^{18}O$ in benthic foraminifera from sediment coresThibodeau et al. (2018), for 1450 – 1780. The second row shows winter JFM, summer JJA, and the annual ANN ensemble mean for the period 1450 – 1780, respectively. The third row shows the standard deviation of AMOC (Sv) for JFM, JJA & the annual averages, respectively. Positive values in the mean represent clockwise circulations.





## 3.2 Atmospheric and surface oceanic patterns in opposite AMOC phases

Winter (JFM) Sea Level Pressure (SLP) composites show the characteristic dipole of the positive NAO pattern during the $HAI_{year}$(Fig. 2b), while the dipole sign flipped when considering $LAI_{year}$ showing a negative NAO pattern (Fig. 2a). The changes from $NAI_{year}$ to either $LAI_{year}$ or $HAI_{year}$ are statistically significant at the 95% interval, especially over the NAO dipole regions. NAO anomalies are associated with the characteristic patterns of air-sea coupling parameters, such as surface wind stress and turbulent heat flux. We calculate turbulent heat flux as the sum of the sensible and latent heat fluxes (Smith and

Polvani, 2021).

For $HAI_{year}$, we find predominantly eastward surface wind stress north of 40°N, and vice-versa below 40°N. $LAI_{year}$ surface wind stress anomalies are relatively less pronounced but with a corresponding westward flow north of 40°N (Fig. 2c & d). Eastward surface wind stress north of 40°N is associated with anomalous turbulent heat flux towards the atmosphere (Smith and Polvani, 2021). This relation between eastward wind stress and heat flux into the atmosphere is evident in our simulations

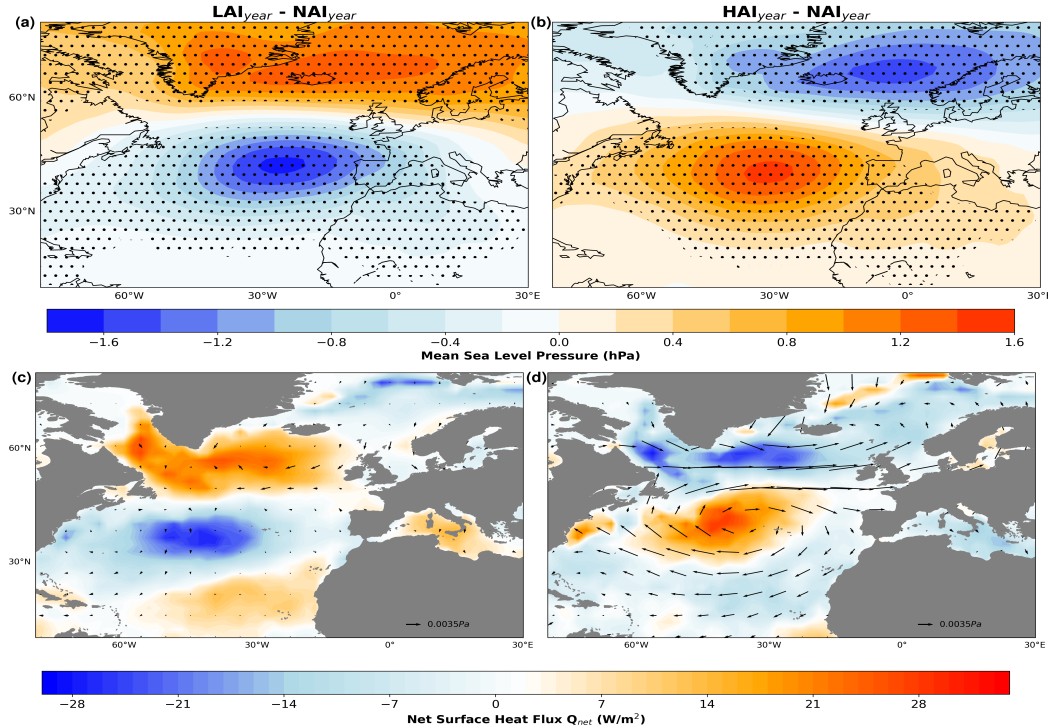

**Figure 2. Winter (JFM) composites of $LAI_{year}$ and $HAI_{year}$ wrt. $NAI_{year}$ for selected atmospheric variables.** The top row shows Mean Sea Level Pressure anomaly (hPa) composites for $LAI_{year}$ (left column) and $HAI_{year}$ (right column). The bottom row shows North Atlantic surface wind stress (vectors) and turbulent heat flux anomaly composites. Stipples represent regions that are statistically significant at 95%. For $Q_{net}$, only statistically significant grid points are displayed, and positive values correspond to heat fluxes into the ocean, while negative values are towards the atmosphere.



with negative $Q_{net}$ in regions of the eastward flow. Conversely, $Q_{net}$ is positive between 30°N and 40°N resulting in a dipole of turbulent heat flux in the North Atlantic. However, there is a switch in this dipole sign when considering the $LAI_{year}$. Interestingly, the $Q_{net}$ pattern resembles the AMOC anomaly for both $LAI_{year}$ and $HAI_{year}$, with negative and positive values around 30°N, respectively (Fig. 3a and b). The signs correspond to the $Q_{net}$ anomaly pattern over the same region. However, the positive/negative stream-function anomalies found in $HAI_{year}$ are highly localized in the Northern Hemisphere (Fig. 3b),

while the negative stream function shown in $LAI_{year}$ extends to the Southern Hemisphere (Fig. 3a). This negative stream function indicates an anticlockwise circulation pattern as a result of weak inflow from the South Atlantic. Weak inflow from the South Atlantic typically disrupts the formation of deep water masses, such as North Atlantic Deep Water (NADW), which are important components of the AMOC. On the other hand, localized positive stream function in $HAI_{year}$ depicts increased or rapid clockwise circulation. This indicates an enhanced deep water formation and suggests an increased production of dense

water in the North Atlantic, which subsequently sinks and contributes to a deep limb of the AMOC during $HAI_{year}$. Winter SST over the North Atlantic shows a predominant statistically significant positive anomaly with few cooling regions during $HAI_{year}$ (Fig. 3d). While positive SST anomalies also prevail during $LAI_{year}$, they are relatively weak and not statistically significant (Fig. 3c). MLD anomalies are largely induced from the atmosphere, through momentum transfer; this is supported by the structures shown in Fig. 2 and 3. Shallow MLD has a reduced thermal capacity thus leading to positive SST anomalies.

Furthermore, MLD at high latitudes is an indicator of deep convection sites, especially in the North Atlantic, which is an indirect measure of the AMOC strength. Our simulations agree with observations (Marshall et al., 1999; Yeagerz et al., 2021) by showing major sites of deep convection over the Labrador Sea, south of Iceland, the Norwegian Seas, and over the Gulf Stream extension into the North Atlantic. Considering differences between the composites of previously selected AMOC index years, MLD over most of the North Atlantic is unchanged. However, there are significant differences in MLD over the

Norwegian Sea and the Gulf Stream extension into the North Atlantic. Our analyses of winter MLD show increased depth over the Norwegian Sea during $HAI_{year}$ (Fig. 3f) while the Gulf Stream extension becomes shallow. On the contrary, $LAI_{year}$ shows shallower MLD over the Norwegian Sea with an accompanying significantly deeper Gulf Stream extension (Fig. 3e). The Labrador Sea is the primary region of convection and deep water formation in the North Atlantic, and it exhibits high internal variability in the simulations. To completely capture all internal variations, a large number of ensemble simulations

either from a single model or in a multi-model approach is required. The atmospheric simulations (ModE-Sim) used as forcing for our stand-alone ocean simulations is a 20-member ensemble, implying a medium-sized ensemble. Furthermore, they use the same SSTs as used for nudging here. Based on these considerations, we select the 10-member SSTs and atmospheric forcing that maximizes the variance of the medium-sized ensemble, thereby covering the range of possible internal variability. Since our simulations are nudged in the surface to annual SST, we also compared the annual composites of SST and MLD

for $LAI_{year}$ and $HAI_{year}$. Composites of SST on annual timescale show a more pronounced cooling during the $LAI_{year}$ and also a more widespread statistical significance (Fig. 4a and b), compared to JFM while during $HAI_{year}$, statistically significant positive anomalies are prevalent. Also, we found statistically significant positive MLD anomalies with respect to $NAI_{year}$ over the Irminger and Labrador Seas (Fig. 4c and d). In contrast, positive significant anomalies are evident over the Norwegian and the Arctic Seas for $HAI_{year}$.





**Figure 3. Winter (JFM) composites of LAI$_{year}$ and HAI$_{year}$ for selected oceanic variables.** The top row shows Mean AMOC anomaly (Sv) composites for LAI$_{year}$ (left column) and HAI$_{year}$ (right column). The middle row shows SST (K) for LAI$_{year}$ (left column) and HAI$_{year}$ (right column) while the bottom row shows the same but for North Atlantic Mixed layer depth (m). Stipples represent regions that are statistically significant at 95%.

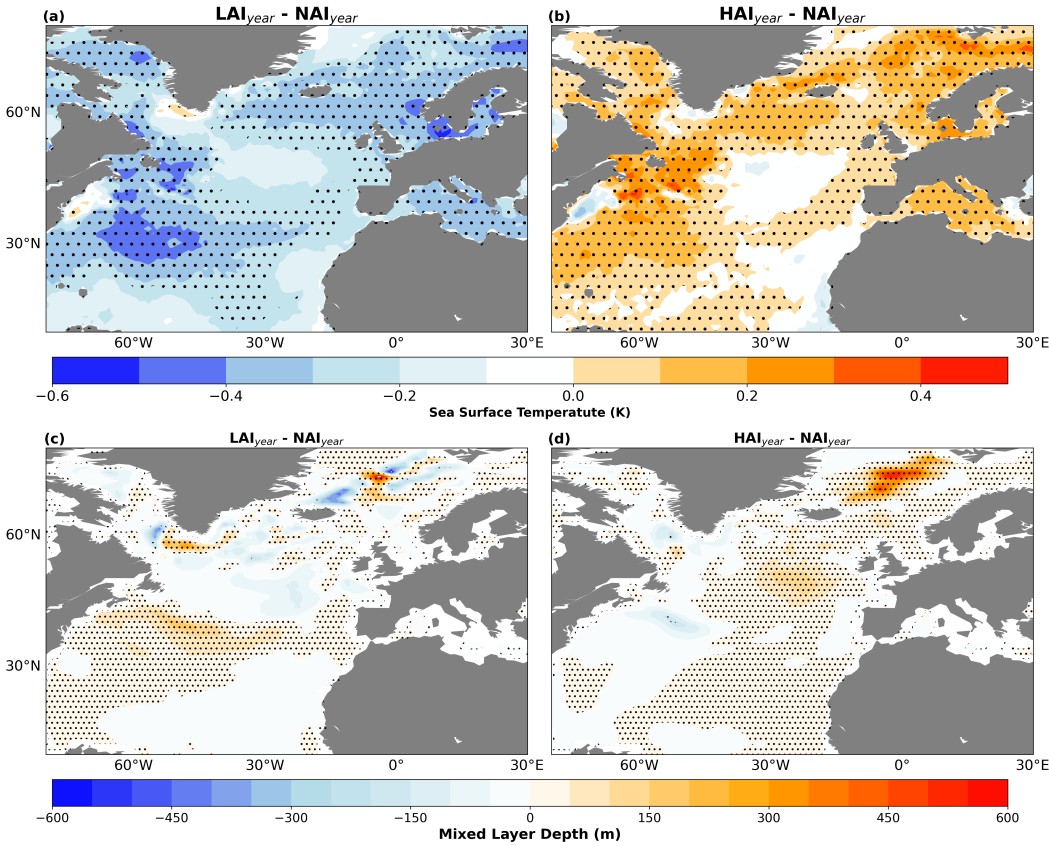

**Figure 4. Annual mean composites of LAI$_{year}$ and HAI$_{year}$ for selected oceanic variables.** The top row shows the annual mean North Atlantic SST anomaly (K) for LAI$_{year}$ (left column) and HAI$_{year}$ (right column) while the bottom row shows the same but for the North Atlantic Mixed layer depth (m). Stipples represent regions that are statistically significant at 95%.

## 3.3 Atmospheric forcing of the AMOC Variability

To further investigate the relationship between the AMOC and atmospheric variables, we compute the NAO index using SLP from the ECHAM6 variables used as forcing, for the entire period and regress it on the spatially complete AMOC fields in different lag-time. We calculate the NAO index as the first principal component of SLP in the North Atlantic sector (Fig. S5). These PC-based indices of the NAO are the time series of the leading Empirical Orthogonal Function (EOF) mode of SLP anomalies over the Atlantic sector, 20° - 80°N, 90°W - 40°E (Hurrell et al., 1995). The lagged overturning response during the Little Ice Age is significant at around 30°N, with the maximum regression coefficient of 0.16 Sv/hPa extending from the surface to 5 km depth (Fig. 5c, d, e & f). Thus, the AMOC variability is closely linked to the NAO under stable CO$_2$ conditions. This positive response generally resembles the composite pattern of the AMOC during HAI$_{year}$ (Fig. 3). In contrast, the negative response dominant from 20°N into the Southern Hemisphere is a characteristic of the LAI$_{year}$ AMOC pattern. Furthermore,





the positive regression coefficient of the overturning stream function to the NAO index appears persistent in our simulation and gradually reduces in intensity as the lag time increases from -1 to -10.

We also examine the lead-lag correlations between the NAO index and the estimated AMOC index at 30°N, using the ensemble mean of our simulations. The correlation curve between NAO and AMOC is based on decadally smoothed and unfiltered data (Fig. 5a). The lag correlations between the NAO and AMOC (Fig. 5a) peak at lag zero for the unfiltered, and we find significant

positive correlations (0.58 for unfiltered data and 0.52 for decadally smoothed). The positive correlations are statistically significant at a 95% confidence interval when the NAO leads the AMOC and gradually decreases up to 10 years in the decadally smoothed analyses, while the corresponding lead year decreases between 2 to 5 years in the unfiltered. Similarly, the maximum correlation coefficients using each ensemble member of our simulations show positive correlations, ranging from 0.3 to 0.7 (Fig. 5b), indicating that the NAO largely impacts the interannual to decadal variability of the AMOC between 1450 - 1780 CE

(Fig. S6).

Besides, the thermohaline forcing and high-frequency wind variability strongly influence the interannual to decadal variability of the AMOC over the mid-latitudes of the North Atlantic (Biastoch et al., 2008; Song et al., 2019). Therefore, we regress the NAO index on the turbulent heat flux ($Q_{net}$, JFM) and surface wind stress for $HAI_{year}$ and $LAI_{year}$, separately to depict the NAO-induced pattern of air-sea coupling parameter during the $LAI_{year}$ (Fig. 6a) and $HAI_{year}$ (Fig. 6b). These patterns

provide a baseline to determine the time-dependence of the AMOC on $Q_{net}$ and surface wind stress, considering $HAI_{year}$ and $LAI_{year}$, separately. Additionally, we regress the selected $HAI_{year}$ and $LAI_{year}$ AMOC index (calculated at 26.5°N) on the $Q_{net}$ and surface wind stress using different time lags.

 The resulting patterns for $LAI_{year}$ are typically those that are linked to the NAO (Wen et al., 2005; Visbeck et al., 2013) with westward surface wind stress between 40° and 60°N and eastward surface wind stress south of 30°N (Fig. 6a). The westward

flow of surface wind stress is accompanied by reduced surface heat flux around 30°N and between 30° - 60°W, i.e., a negative heat flux towards the atmosphere. Similarly, for $HAI_{year}$, westward surface wind stress prevails between 40° and 60°N, while eastward surface wind stress is found between 20° and 40°N (Fig. 6b). Negative heat flux is also identified for $LAI_{year}$, but its spatial extent is increased compared to the one for $HAI_{year}$. Furthermore, negative heat flux prevails in the Greenland Sea and extends toward the Norwegian Sea.

A similar regression pattern between the AMOC index and turbulent heat fluxes $Q_{net}$ for the $LAI_{year}$ is found when turbulent heat flux leads the AMOC by 4 to 7 years (Fig. 7). The negative regression coefficients between the AMOC index and the heat flux, found in the $LAI_{year}$, decrease the meridional temperature gradient in the subpolar North Atlantic from Lag -3 to -7 years. This indicates a strong link between surface heat flux and the slowdown of the AMOC on a timescale of 4 to 7 years. A prominent feature in the $LAI_{year}$ analyses is the surface heat gain over both the Irminger Sea and the Labrador Sea (Fig.

6). The Irminger and the Labrador Sea are reported to be possible sources of AMOC variability (Megann et al., 2021; Chafik et al., 2022). Surface heat gain seen over the Irminger and Labrador Sea insulates the ocean surface through the lack of deep convection and thus weakens the AMOC (Lohmann et al., 2021). Our results show that this heat loss south of these regions is not sufficient to sustain the AMOC circulation, hence a weak phase (Fig. 6).







**Figure 5. Lead–lag correlation and regressions of AMOC index and North Atlantic Oscillation NAO.** The top row shows Lead–lag correlation between the ensemble mean of the NAO index and AMOC index at 30°N, calculated from our 10-member simulations. The red (blue) line is for the original annual (decadally smoothed) time series. Negative (positive) lags indicate that the NAO is leading (lagging), and the red (blue) dashed lines are the 95% confidence levels for the unfiltered (smoothed) time series based on the effective numbers of degrees of freedom. The histogram shows the correlation distribution of all ensemble members. The bottom row shows the Lag-regression of the normalized NAO index on the Atlantic meridional overturning stream function. Stippling indicates significance at the 95% confidence interval.



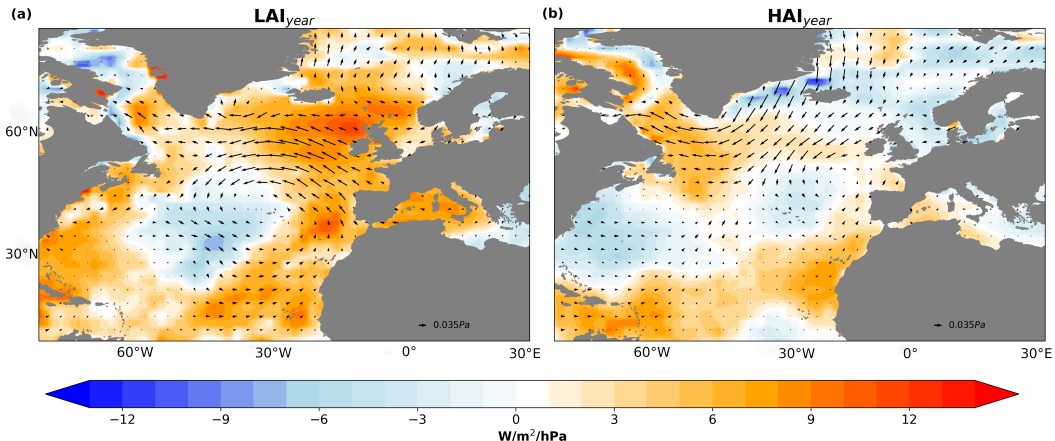

**Figure 6. Regression of LAI$_{year}$ and HAI$_{year}$ NAO index onto the North Atlantic surface wind stress (vectors) and turbulent heat fluxes** $Q_{net}$. For $Q_{net}$, positive values correspond to heat fluxes into the ocean, while negative values are towards the atmosphere. For Q$_{net}$, only statistically significant grid points are displayed.

For HAI$_{year}$, wind stress tends to play a more crucial role. The lag-regression of surface heat flux onto AMOC index on timescales of 1 to 7 years (Fig. 7) shows a contrasting pattern to the regression coefficients of the HAI$_{year}$ heat flux onto NAO index (Fig. 6). However, the wind stress pattern shows the same for the 1-year time lag (Fig. 7), indicating an instantaneous response of the AMOC to surface wind stress during HAI$_{year}$. The result suggests an enhanced cold air flow from the Arctic into the Labrador Sea (Fig. 6). This cold air flow typically promotes surface heat loss in the Subpolar North Atlantic, providing

the ground for increased upwelling, thereby intensifying the strength of the AMOC (Vic et al., 2019; Lohmann et al., 2021). To further confirm the mechanisms influencing the different phases of the AMOC, we compute lead-lag regression of MLD onto NAO and AMOC index. Our results show widespread statistically significant positive MLD regression coefficients over the entire North Atlantic with a few patches of negative coefficients (Fig. 8a and b). The positive regression coefficients are pronounced over the Irminger, Labrador, Greenland, Norwegian, and Arctic Seas. This establishes a baseline for the time

dependence of the NAO-influenced component of the AMOC. The regression of the AMOC index on the different AMOC phases shows statistically significant positive regression coefficients over the Irminger and the Labrador Seas from lag -4, and the spatial extent gradually decreases until lag -7 for LAI (Fig. 9), while during the strong AMOC phase (Fig. 10), widespread positive regression coefficients are evident in the North Atlantic on a timescale of lag -1 to -4 and gradually reduces from -5 to -7. Furthermore, there is a dipole of positive and negative coefficients over the Arctic and the Greenland Seas. This corresponds

to the region of flow from the Arctic to the Labrador Sea. Our result here shows the influence of the Irminger and the Labrador Sea on the weak phases of the AMOC, while the Arctic, Greenland, and Labrador Seas all contribute to the strong phase of the AMOC variability.



**Figure 7. Regression of LAI$_{year}$ and HAI$_{year}$ normalized AMOC index onto selected atmospheric variables.** Shown for LAI$_{year}$ and HAI$_{year}$ is the North Atlantic surface wind stress (vectors) and turbulent heat flux regression. For Qnet, positive values correspond to heat fluxes into the ocean, while negative values are towards the atmosphere. Negative lags indicate that heat flux is leading. For Q$_{net}$, only statistically significant grid points are displayed.



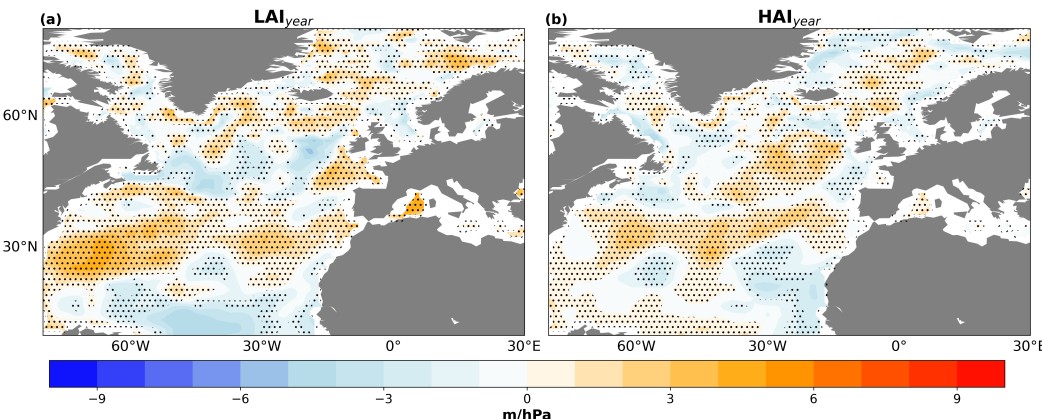

**Figure 8. Regression of $LAI_{year}$ and $HAI_{year}$ NAO index onto the North Atlantic Mixed Layer Depth.** Shown are the regression coefficients of NAO index on MLD during $LAI_{year}$ (a) and $HAI_{year}$ (b). Stipples represent regions that are statistically significant at 95%.

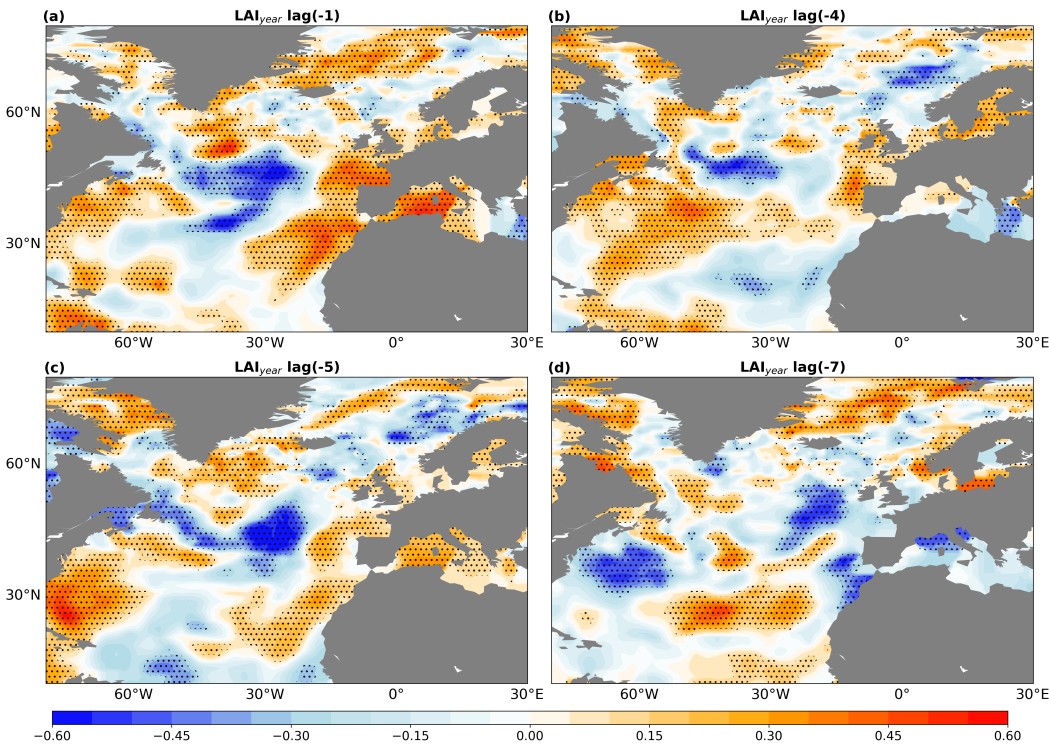

**Figure 9. Standardized regression coefficients of $LAI_{year}$ AMOC index onto corresponding mixed layer depth for different time lags.** Shown are the regression coefficients at a lag time of 1 year (a), 4 years (b), 5 years (c) and 7 years (d). Stipples represent regions that are statistically significant at 95%.




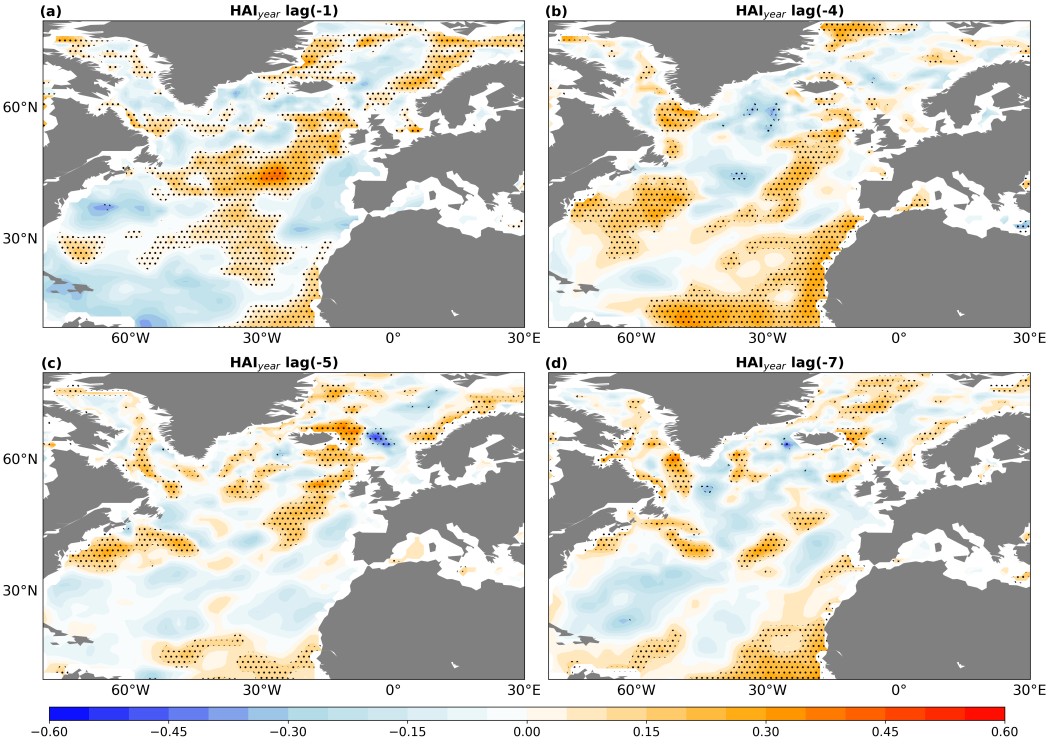

**Figure 10. Standardized regression coefficients of HAI$_{year}$ normalized AMOC index onto corresponding mixed layer depth for different time lags.** Shown are the regression coefficients at a lag time of 1 year (a), 4 years (b), 5 years (c) and 7 years (d). Stipples represent regions that are statistically significant at 95%.

## 4  Discussion

Here, we present a new modeling approach to reconstruct the AMOC from 1450 to 1780 CE, utilizing an ensemble of stand-
alone ocean model simulations nudged to proxy reconstructed SSTs. We used this reconstruction to assess various known
mechanisms triggering the different phases of the AMOC under stable atmospheric CO$_2$ concentration.

The new reconstruction of the AMOC is consistent with previous studies (Yoshimori et al., 2010; Jungclaus et al., 2013) in terms
of mean state and variability. Furthermore, it agrees with reconstructions of the AMOC strength using paleoclimate proxies
(Thibodeau et al., 2018). Although reconstructions may be biased in the representation of some AMOC components than
others, our results show the robustness of the modeling approach in simulating AMOC states that are reasonably comparable
with other existing evidence and thus can make further inferences about AMOC variability for the period 1450 - 1780 CE.

Using the new reconstruction, our results confirm existing model-based hypotheses of the multi-annual AMOC variability
under unforced (compared to present-day) climate conditions, in particular the role of the NAO (Raible et al., 2001; Song et
al., 2019). We find that AMOC variability from 1450 to 1780 CE is in phase with the NAO in winter, as suggested by several
studies (Biastoch et al., 2008; Sun et al., 2015; Wen et al., 2016). The in-phase relationship between the AMOC and winter



NAO highlights the close connection between the ocean and atmosphere, and the importance of understanding their interactions in order to better predict changes in the climate system. Our composite analyses of SLP show that positive and negative NAO prevail during strong and weak phases of the AMOC, respectively (Fig. 2). Furthermore, the relationship between the AMOC and NAO in winter is influenced by air-sea coupling parameters such as wind stress, and turbulent heat fluxes. During a positive

NAO phase, the winds over the North Atlantic tend to blow from the southwest, creating a positive wind stress curl that drives the AMOC and warms the North Atlantic, warm SSTs are favored in the western North Atlantic, while cold SSTs are favored in the eastern North Atlantic, creating a strong SST gradient that can drive the AMOC. During a negative NAO phase, the winds tend to blow from the northwest, creating a negative wind stress curl, which weakens the AMOC and cools the North Atlantic (Zhang et al., 2005).

Also, we show the effect of surface heat gain over both the Irminger Sea and the Labrador Sea in insulating the ocean surface through the lack of deep convection and thus weakening the AMOC. Heat loss and gain over the Irminger Sea and Labrador Sea have been identified for influencing variability of the AMOC (Lohmann et al., 2021; Megann et al., 2021; Chafik et al., 2022), by controlling the density of water in these important deep water formation sites. However, some coupled climate models have difficulty reproducing the observed deep water formation rates and patterns in the North Atlantic. This is due to the limited

representation of ocean-atmosphere interactions in these models resulting in inaccurate simulations of deep water formation (Piron et al., 2017). Similarly, the prescription of boundary conditions in these models could influence the locations for density transformation thereby affecting the water mass properties (Rüh et al., 2021). This leads to the underestimation of the AMOC variability and its amplitude (Ortega et al., 2017). Our nudging technique largely corrects for this deep water formation deficit by using realistic SST from paleoclimate proxies. One drawback of the nudging approach is the use of a stand-alone ocean

model minimizing ocean-atmosphere feedback. This lack of atmospheric-ocean coupling may cause the simulated AMOC to be sensitive to certain variables such as heat fluxes (Kostov et al., 2019). In coupled climate model simulations, the ocean typically experiences feedback from atmospheric circulation that adjust to changes in SST (Msadek et al., 2009). Still, the ensemble nudging approach allows the simulation of the LIA ocean by constraining the deep ocean state and enables the analysis of its accompanying atmospheric and surface ocean variability.

## 5   Summary and conclusions

In this study, we nudged a stand-alone ocean model MPI-OM to proxy-reconstructed SST. Based on these model simulations, we introduce new estimates of the AMOC variations during the period 1450 - 1780 through a 10-member ensemble simulation with a novel nudging technique. Furthermore, our approach reaffirms the known mechanisms of AMOC variability and also improves existing knowledge of the interplay between the AMOC and the NAO during the AMOC's weak and strong phases.

We found a lag-correlations between the NAO and AMOC that peaks at lag zero, and are statistically significant at a 95% confidence interval when the NAO leads the AMOC. Also, our results suggest that the AMOC weak phases under stable atmospheric $CO_2$ conditions result from a 4-to-7-year lagged effect of surface heat flux associated with the NAO. In contrast, strong phases are a response to instantaneous surface wind stress. Our study shows the distinctive impact of key oceanic sec-





tors on AMOC's weak and strong phases. The Irminger and Labrador Seas dominate during the weak phase, while the Arctic,

Greenland, and Labrador Seas drive the strong phases. We find a reduction in deep convection driven by surface heat gain over Irminger and Labrador Seas. This coherent pattern aligns with observed MLD anomalies in lead-lag regressions with NAO and AMOC indices. These findings highlight the asymmetry of the drivers of the AMOC's weak and strong phases.

Furthermore, determining the strength of the AMOC beyond the instrumental period relies on paleoclimate proxies, which are indirect measures of past oceanic and atmospheric conditions. However, the relevance of different paleoclimate proxies to the

AMOC varies, and some locations are better suited to capturing specific aspects of the ocean and atmosphere. The limited scope of data available from these proxies, combined with uncertainties and conflicting evidence (Caesar et al., 2018; Latif et al., 2022), pose challenges in drawing definitive conclusions about whether the AMOC is currently undergoing an unprecedented weakening (Latif et al., 2022; Kilbourne et al., 2022). Our findings highlight the implementation and importance of nudging a stand-alone ocean model to a realistic proxy reconstructed SST can help constrain past AMOC variability. Finally,

it delivers a new baseline to place recent changes in the AMOC in a long-term context, i.e., the more rapid slowdown in the present day compared to the preindustrial period.



*Code availability.* Codes to produce the figures are available from the corresponding author. A modified MPI-OM run script can be found on GitHub: https://github.com/shamakson/Ocean_modelling_stuffs.

*Data availability.* Simulated spatially complete 10-member ensemble AMOC dataset used in this study are available at https://doi.org/10.5281/zenodo.7648938.

*Author contributions.* ES and SB developed the study framework. ES carried out the SST reconstructions, ocean model simulations, and analyses, with inputs from CR, RH, AF, and SB. Finally, ES wrote the manuscript implementing comments from all co-authors.

*Competing interests.* The authors declare that there is no conflict of interest.

*Acknowledgements.* This project has received funding from the European Research Council (ERC) under the European Union's Horizon 2020 research and innovation programme grant agreement No. 787574 "PALAEO-RA". ECHAM6 and MPI-OM simulations were performed at the Swiss Supercomputer Centre (CSCS). CCR received support from the Swiss National Science Foundation (grant no. IZCOZ0_205416). ARF was funded by the European Union's Horizon 2020 research and innovation programme under the Marie Skłodowska-Curie grant No. 894064 (AQUATIC).



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
