# Peer review of "Multi-annual variability of a new proxy-constrained modeled AMOC from 1450-1780 CE"

_Climate of the Past, 2023_

## Referee Comment (RC2)

This paper proposes to reconstruct the AMOC over part of the last millennium. The method chosen is to use an ocean-only model simulation forced by both atmospheric forcing from a coupled model simulation over the last millennium and where a surface nudging towards reconstructed SST is also included. The authors then claim that their reconstruction of the AMOC is robust and in agreement with other reconstructions, and then go into an analysis of the influence of the NAO on the behavior of their reconstructed AMOC.

There are several serious issues with this paper, in terms of experimental design, statistical analysis, physical understanding and claim and conclusions from poor evidences. Those issues are so serious, that I should say that a great number of the conclusions of this article are mostly flawed.

Here are the main issues I have:
- The experimental design is questionable: why using ocean-only simulation, when we know that ocean and atmosphere are strongly interacting for what concerns the AMOC variability. Using ocean-only model might strongly affect the variability of the AMOC, since in this ocean-only model configuration, the SSS is restored towards climatological observations. This is a very strong choice, since salinity is playing a key role in a number of models for what concern AMOC variability (e.g. Menary et al. 2015). This strong choice is almost not discussed in the paper, and the word salinity, a well-known key variable for the AMOC since decades, is just cited once in the paper (to say that it is restored). When accounting for this strong hypothesis, it is quite clear that what remains as an external (to the ocean) driver of the AMOC is the heat or momentum flux, largely driven by NAO in many models. There is not much novelty here I should say
- This leads to my second main points: most of the results depicted here are focused on the impact of the NAO on the AMOC. There is a huge literature on this topic (a few of it is cited in the paper). What is new in this paper is not really assessed and we are left with a long analysis of the impact of the NAO on the AMOC in a stand-alone ocean model, as it was used to be done 20 years ago. The model is coarse resolution, so this is far from clear if what is depicted is representative of any real dynamics. Also the main correlation between AMOC and NAO is found in phase (except when smoothing the data, where a lag appears, but the significance is not really properly assessed as far as I can see, which might be a nice example of how smoothing can lead to unrobust results). It is unclear how the authors are determining the fact that the NAO is therefore leading. With a zero-lag correlation, we are clearly facing a chicken and egg problem. Also Fig. 5 is poorly depicted by its caption and there is a missing red dotted line, and we are left aside to know how the effective number of freedom has been computed for the smoothed lines.
- Also, this long description of some potential mechanism linking the AMOC and the NAO is totally losing the main point of the paper which was to reconstruct the AMOC over part of the last millennium. On this aspect I have very serious doubt, since the restoring towards reconstruction is only concerning the SST. The Ortega et al. (2017) paper cited by the authors is actually showing that SSS nudging is a prerequisite to be able to reconstruct the AMOC, as it was already established in Servonnat et al. (2015). SST alone is usually not very efficient to reconstruct very well the AMOC in an AOGCM (cf. Fig. 10 from Servonnat et al. 2015). For very specific conditions, for instance

following a volcanic eruption, it can work (cf. Swingedouw et al. 2013). But over such a long period, we can have some serious doubts.

- Also, something that is totally left appart in the discussion of the mechanism concern the role played by the nudging. The SST in the simulations analysed are driven by surface fluxes, advection, diffusion AND the nudging term that is added. This term is nowhere shown or discussed. It is simply assumed that it will allow to reconstruct the AMOC (while previous points show it is likely not true…)
- Another crucial point is that the validation of the reconstruction is made very rapidly with vague statement like "consistent", or correlation number where significance is even not assessed (comparison with Rahmstorf et al. (2015). I urge the author to correctly assess the "consistence" of their reconstruction with other published one, using quantified metric, with a proper estimation of the significance (accounting for the auto-correlation of the time series in the degrees of freedom when assessing significance of the correlation at the very least). Other reconstructions are also now available and might be of interest, e.g. Michel et al. (2022) for the AMV, strongly linked to the AMOC in models.
- Lastly, we can wonder why is the reconstruction only proposed over 1450-1780. This is not that well explained I think, and give the impression that weird things are happening for the rest of the last millennium

Following all those key issues, I cannot recommend this article for publication without a very substantial work on the robustness of the results, the logic of the experimental design, the assessment of the novelty regarding existing literature, etc. I therefore propose to reject this paper, although I see some potential interest in what has been done, but deserve a lot of additional work to be consistent, robust and allow the writing of a clear paper (whose conclusion are based on evidence, and the questions are clearer).

**Bibliography**

Menary, M. B., D. L. R. Hodson, J. I. Robson, R. T. Sutton, R. A. Wood, and J. A. Hunt (2015), Exploring the impact of CMIP5 model biases on the simulation of North Atlantic decadal variability. Geophys. Res. Lett., 42, 5926–5934, doi:10.1002/2015GL064360.

Michel S., Swingedouw D., Mignot J., Gastineau G., Ortega P., Khodri M., McCarthy G. (2022) Early warning signal for a tipping point suggested by a millennial Atlantic Multidecadal Variability reconstruction. *Nature Communications* 13, 5176 (2022).

Servonnat J., Mignot J., Guilyardi E., Swingedouw D., Seferian R., Labetoulle S. (2015) Reconstructing the subsurface ocean decadal variability using surface nudging in a perfect model framework. *Climate Dynamics*, 44 (1-2), 315-338, DOI 10.1007/s00382-014-2184-7.

Swingedouw D., Mignot J., Labetoule S., Guilyardi E. and Madec G. (2013) Initialisation and predictability of the AMOC over the last 50 years in a climate model. *Climate Dynamics* 40, 2381-2399. DOI: 10.1007/s00382-012-1516-8